# BACKOFF DECODING: AN INFERENCE ACCELERATION FRAMEWORK FOR LANGUAGE MODELS WITH A TUNABLE EFFICIENCY-PERFORMANCE TRADEOFF

## ABSTRACT

In current transformer-based language models, all tokens in a sequence are generated by identical forward passes and thereby incur the same inference cost. However, tokens vary widely in their importance to the overall generation and their difficulty for models to generate correctly, making this equal allocation of inference resources suboptimal. We introduce backoff decoding, a framework for efficient language model inference that dynamically allocates token generations between two (or more) models of different sizes, according to an arbitrary decision function. By modifying how this decision function allocates generations between the differently sized models, users can tune their generation along an efficiency-performance tradeoff to suit the needs of their application. Backoff decoding can be used on any set of models with the same tokenizer and does not require any training or finetuning of the models themselves. As a demonstration of our framework, we show that backoff decoding with a large and a small model can significantly reduce inference cost while sacrificing virtually no performance compared to the standalone large model. We then show that inference costs can be reduced even further, achieving inference accelerations of up to 3-4x in exchange for reductions in model performance, demonstrating an efficiency-performance tunability not found in other inference acceleration techniques.

## 1 INTRODUCTION

Transformer-based language models have demonstrated impressive generation capabilities across a variety of complex tasks (Brown et al. (2020); Hendrycks et al. (2021); Chen et al. (2021)) and thus have found applications in numerous real-world scenarios. The performance of these models is generally known to scale with their size and thereby their inference cost, facing users with a inference cost-performance tradeoff when deciding which model size to use (Kaplan et al. (2020)). However, these models generate tokens autoregressively by passing inputs through a full, identical forward pass during each token generation, and thus split their inference costs evenly across all generated tokens. This means that when using a larger model, users incur the higher inference cost across all tokens uniformly, irrespective of the nature of each individual token.

Given that tokens vary widely in their importance to the overall sequence and their difficulty to generate correctly, this is a suboptimal allocation of inference resources, either forcing users to incur unnecessary inference costs or forcing them to forgo substantial performance improvements. Firstly, not all tokens have the same importance in terms of determining the quality or meaning of the final output sequence. For instance when generating answers to multiple choice questions, the token indicating the answer choice (e.g. A, B, C, D) is significantly more important than the surrounding tokens. Likewise, not all tokens have the same generation difficulty: when generating the sequence 'AI researcher Geoffrey Hinton studied at the University of Toronto', the tokens corresponding to 'Hinton' given prefix 'AI researcher Geoffrey ' are considerably easier to generate than the tokens corresponding to 'Toronto', since the latter require an understanding of the subject of the sentence as well as the parametric knowledge of where Geoffrey Hinton studied. Even a simple n-gram model might be able to correctly generate the tokens for 'Hinton' in the first case, while the second likely requires a well trained, complex language model to be generated correctly. However, current language models will spend the same inference resources on both cases.

This same phenomenon can be illustrated through the example of generating a single token answer to a multiple choice questions. If a much smaller model would already generate the correct answer to a given question, using a larger model to answer the same question would be a waste of the additional compute, as we could have used the more efficient small model and achieved the same outcome. On the other hand, it would be an effective use of additional compute to use the large model on questions the small model would get wrong, as in this case the additional compute would result in an improved outcome. Again, current language models do not account for this difference in difficulty, and will use the same inference resources across both cases. If users want to use the larger model to perform well on the questions the smaller model cannot handle, they must accept a higher inference cross across all tokens.

Given this flaw of current generation techniques, it would be a significant improvement to inference efficiency to dynamically determine how much compute a given token will need and allocate inference resources accordingly. Furthermore, a framework that dynamically allocates inference resources would give users the freedom to tune this allocation to different points along an inference cost-performance trade-off depending on the specific needs of their application - an option which current inference acceleration techniques do not allow.

Thus, we introduce **backoff decoding**, a tunable framework for efficient language model inference that dynamically allocates token generations between two (or more) models of different sizes, according to an arbitrary decision criteria. Our approach can be used on any set of models with the same tokenizer, and thus can be applied to virtually all models from common model families. Our approach does not require any finetuning or training of the models themselves, and only requires training of a classifier or other decision mechanism to effectively allocate generations between models. To demonstrate our approach, we show that backoff decoding with a large and a small model can result in substantial decreases in inference cost without sacrificing any performance compared to the large model. We also show that users can reduce inference costs even further way past the level of current SOTA inference acceleration techniques, achieving inference accelerations of 5-6x, at the cost of just a small decrease in overall performance, demonstrating the efficiency-performance tunability of our framework.

## 2 RELATED WORKS

The idea of dynamic inference resource allocation, sometimes also called 'adaptive computation methods', has been previously explored in several works (Han et al. (2021); Sukhbaatar et al. (2019); Schwartz et al. (2020)), most of which focus on early stopping methods (Schuster et al. (2021); Scardapane et al. (2020); Bapna et al. (2020); Elbayad et al. (2020)). These approaches typically operate off of a single model and define output heads over several or all of the hidden states of the model. During inference, once a given confidence threshold or other decision criteria is achieved, these models will stop their forward pass early, producing outputs using the hidden state and corresponding output head at the given layer and thereby dynamically allocate the inference resources spent on each inference call.

While they have demonstrated some promising results, these early stopping methods have two major flaws. Most notably, they do not work on out-of-the-box language models and typically require extensive training or finetuning in order to be properly optimized for this early-stopping objective. This significantly reduces the applicability of these methods, as users must implement and train these methods themselves. Furthermore, these methods have not yet been able to achieve state-of-the-art language modeling performance and are much weaker compared to the strong performance of standard transformer model. It's unclear whether this lower performance is due to the early stopping mechanism itself or simply a consequence of it little research in this area so far. However, it is possible that the early stopping objectives complicates the learning dynamics of the model during training, since each layer is being optimized to play a different role during inference depending on when the model exits.

While not directly an adaptive computation method, another similar approach is speculative decoding (Chen et al. (2023); Leviathan et al. (2023)), which uses a smaller draft model to speculate on potential future token sequences and has the large model accept or reject these small model generations. Just like backoff decoding, this methods does effectively leverage the difference in inference cost and performance between models of different sizes and thus results in a substantial inference

cost reduction. However, since the large model must still verify all token generations, speculative decoding does not allow users any tunability of the inference cost-performance tradeoff. In other words, after realizing the inference cost reductions from speculative decoding, users do not have the option to further lower inference costs in exchange for reductions in performance - an option which would find several useful applications in real world implementations.

Backoff decoding addresses the major concerns of both of these methods. Firstly, it does not require any training or finetuning of the models themselves and works out-of-the-box on any set of models with the same tokenizer. As such, it also only uses seperately trained models, and therefore does not have an issue with conflicting learning objectives during training introduced by leveraging different model sizes. Most importantly, backoff decoding gives users the flexibility to tune the efficiency-performance tradeoff of their model's generation, allowing them to optimize for different objectives depending on the needs of their application.

## 3 THE BACKOFF DECODING FRAMEWORK

Consider two language models, a large model $M_L$ and a small model $M_S$, both with the same tokenizer $T$. Autoregressive LMs produce a distribution over the next token given a sequence of preceding tokens, which is described by a logits vector, i.e., $M_{L/S}(x_1, \ldots, x_{t-1}) \in \mathbb{R}^V$. Given the identical tokenizers, these models can be used interchangeably at any step of an autoregressive generation. Due to their different sizes, generating all tokens with $M_S$ will result in the fastest generation time but with the worst performance, while generating all tokens with $M_L$ will result in the slowest generation time but with the best performance. Between these two extremes lies an inference cost-performance tradeoff defined by the proportion of tokens generated by $M_S$ compared to $M_L$.

Now consider a decision function $f_d$, which at each generation step determines whether to use $M_L$ or $M_S$ to generate the logits for the next token. This decision function can be any arbitrary function that optimizes for any objective, depending on the application context. Thus, at a given generation step, the logits output by this general backoff decoding model $B_g$ are given by:

$$B_g(X) = \begin{cases} M_L(X) & \text{if } f_d(D) = 0, \\ M_S(X) & \text{if } f_d(D) = 1, \end{cases}$$

where $X = (x_1, \ldots, x_{t-1})$ is the sequence of preceding tokens, $D$ is the arbitrary decision function input and $f_d$ is the decision function.

On the surface, it might initially seem unclear why increasing the number of generations routed to $M_S$ decreases the overall inference cost, since each generation from $M_L$ must still recompute the key-value pairs (and thereby hidden states) for all past sequence positions generated by $M_S$. However, the key here is that if $M_S$ has generated a series of tokens, the computation of the key-value pairs for these tokens in $M_L$ can be done in parallel as opposed to sequentially, resulting in the inference cost reductions we describe. Thus, as long as sufficient tokens are generated by $M_S$ in series, the runtime reductions achieved passing past tokens through $M_L$ in parallel will outweigh the incremental inference cost increase of computing the key-value pairs for all positions in both $M_S$ and $M_L$, resulting in an overall reduction in runtime that grows with the number of tokens generated by $M_S$. This is very similar to how speculative decoding achieves its efficiency gains (Chen et al. (2023); Leviathan et al. (2023)). By having the faster draft model sequentially generate potential token sequences, speculative decoding can use the large model to verify these generations in parallel, leading to inference cost reductions in the case that enough draft model speculations are accepted by the large model. The major difference in backoff decoding is that we always 'accept' the generations from $M_S$, instead relying on the decision function keep the distribution we are sampling from similar to that of the large model.

Given the separate and interchangeable parts of this framework, this general backoff model $B_g$ is highly flexible to context-specific modifications. The framework can easily be extended by changing the number and types of models, as well as modifying the decision function and it's objective to suit a variety of applications. In this work, we will focus on a simple two model backoff decoding framework with `Llama-3.1 8B Instruct` as $M_S$ and `Llama-3.1 70B Instruct` as $M_L$

(Dubey et al. (2024)). We will consider the decision function $f_d$ to be a binary classifier defined over the preceding token sequence (i.e. $D = X$), with the objective of maximizing model performance under sample based decoding while routing as many generations to $M_S$ as possible.

## 3.1 Decision Function Setup and Training

Given our objective of optimizing performance under sample based decoding, the goal of our decision function $f_d$ is to identify the generations for which the next word distributions of $M_S$ and $M_L$ will be similar, as well as those for which the distributions will differ. By doing so, we can route the similar generation to $M_S$ while using $M_L$ for the generations for which the distributions are different, decreasing the inference cost by using $M_S$ while minimizing the changes to the distributions used to sample each token. We can use the KL divergence as a measure of this similarity, and allocate generations between the two models depending on whether this divergence is greater or less than some preset threshold $T$.

Thus, we introduce our first backoff model variant, the **oracle backoff model**, which allocates generations according to the true KL divergence between the next word distribution of $M_S$ and $M_L$:

$$B_{\text{oracle}}(X) = \begin{cases} M_L(X) & \text{if } f_o(X) = D_{\text{KL}}(M_L(X) \parallel M_S(X)) \geq T, \\ M_S(X) & \text{if } f_o(X) = D_{\text{KL}}(M_L(X) \parallel M_S(X)) < T, \end{cases}$$

In practice, using the true KL divergence between the two model's distributions as our decision function is infeasible, as it requires us to run both models at every generation step to determine the true KL divergence between the two distributions. This means that we are running both $M_S$ and $M_L$ sequentially, and are therefore no longer realizing any efficiency gains by parallelizing $M_L$ across past $M_S$ generations.

Therefore, we will modify this oracle decision function $f_o$ to instead estimate the KL divergence between the model's distributions, given the input tokens sequence $X$, without running both models. In this work, we have chosen to do this by defining a neural binary classifier over the hidden state of the small model $M_S$. We do this so that we only have to run $M_S$ at every generation step sequentially, and can parallelize $M_L$ across past generations as desired. Thus, we introduce our next backoff model variant, the **classifier-based backoff model**:

$$B(X) = \begin{cases} M_L(X) & \text{if } f_n(X) \geq T, \\ M_S(X) & \text{if } f_n(X) < T, \end{cases} \quad \text{where } f_n(X) = \sigma(\text{MLP}(n\text{th hidden layer of } M_S(X)))$$

Here $\sigma$ is a sigmoid function applied over the last layer output of the MLP. In light of this classifier-based variant and the impracticality of implementing an efficient oracle decision function, it's important to note that this oracle decision function $f_o$ is still a crucial baseline to start with, as it lets us determine how effective it is to allocate generations based on the KL divergence. As such, the oracle model gives us a theoretical upper performance bound for all decision functions based on estimating the KL divergence (as it represents the performance under optimal KL divergence based routing), and thereby demonstrate how improvements to these decision functions will increase overall performance of the model.

In order to train $f_n$, we opted to frame the optimization as a binary classification problem. During training, we did not update the weights of $M_S$ so that we could reuse the same instance of $M_S$ we used for classification for the generation of tokens. Thus, we trained only the MLP of $f_n$ to classify points into a group with KL divergences below threshold $T_{\text{KL}}$ (to route to $M_S$), and a group with KL divergences above threshold $T_{\text{KL}}$ (to route to $M_L$). We pretrained this classifier on a dataset of (input token sequence, KL divergence) tuples generated from the wikitext corpus, before finetuning them on a similar dataset generated from a set of instruction tuning text. In order to choose the KL divergence threshold $T_{\text{KL}}$ that split the training data into positive and negative classes, we observed the distribution of KL divergences over the training dataset and chose a threshold such that around 75% of the points fell below $T_{\text{KL}}$. We then sampled points from these two groups to balance the two classes, such the number of points with KL divergences above and below $T_{\text{KL}}$ were equal. We did this because we wanted to train the classifier to be more familiar with high KL divergence points

and know how to classify these correctly, since false positive classifications (incorrectly routing to $M_S$) is much more prohibitive for accuracy than false negatives (incorrectly routing to $M_L$).

## 3.2 BACKOFF DECODING WITH KEY-VALUE CACHING

Most transformer implementations used today rely on key-value (KV) caching in order to optimize their inference. Since the key and value tensors for a given sequence position remain the same regardless which position we are computing attention scores for, the KV pairs for all positions can be cached and reused once computed, avoiding the redundant recomputation of these tensors in future forward passes. During generation, these implementations will typically pass the full prompt through the model, calculating the logits and caching the KV values for these initial positions, before sequentially computing and caching the KV pairs for generated positions as the generation proceeds.

We can implement KV caching within the backoff decoding framework with only a few minor modifications to this caching procedure. Firstly, we must maintain two separate caches - one for $M_S$ and one for $M_L$. The cache for $M_S$ will be updated in the same way it would under regular inference, since we are calling $M_S$ at every generation step regardless of the backoff decision. However, $M_L$ will not be called at every step, therefore whenever it is called, it's cache may not have the KV pairs for all previous sequence positions. Thus, as the generation proceeds, we must keep track of all sequence positions which have not yet been cached by $M_L$, so that when $M_L$ is eventually called, we can correctly update it's cache with the KV pairs for these unseen sequence positions. This does not mean that $M_L$ will not benefit from KV caching, since it can still leverage the cached KV pairs for all positions up to the last time it was called. The $M_L$ cache will simply just be a few sequence positions behind the current generation step (depending on how many tokens were generated from $M_S$ is series), and will be updated every time $M_L$ is called.[1]

## 4 EXPERIMENTAL RESULTS

### 4.1 EXPERIMENTAL SETUP

To demonstrate the efficacy of our method, we must show that backoff decoding can result in meaningful inference cost reductions without substantially degrading the model's performance. To do this, we implemented both the oracle and classifier backoff variants with $M_S$ and $M_L$ described above and evaluated their performance across a set of benchmarks. The benchmarks we choose for this were CommonsenseQA (CSQA) to test for general QA ability (Talmor et al. (2019)), GSM8K to test for technical and mathematical ability (Cobbe et al. (2021)), and ASQA to test long form generation capability (Stelmakh et al. (2023)). We measure answer accuracy for CSQA and GSM8K, and measure QA-F1 and QA-EM for ASQA. In order to encourage longer generations and better test for the impact of backoff decoding on overall generation quality, we evaluated the CSQA and GSM8K benchmarks in a longform chain-of-thought setting. ASQA is by default a longform generation benchmark, so we evaluated on it as is.

In addition to this benchmark performance, we also measured the average generation time per token at different backoff percentages as a proxy for the per token inference cost. Since the inference cost of the backoff model only depends on the percentage of tokens routed to $M_S$ compared to $M_L$, we can compute the inference cost of a given evaluation retrospectively by using the backoff percentage observed during the evaluation and measuring the per token inference cost of the model at this backoff percentage on a smaller sample generation. Using this technique, we are able to calculate the inference cost reductions the oracle would be able to achieve at its respective backoff percentages, even though in practice the oracle cannot lead to efficiency gains. Likewise, we also assume that the inference cost of speculative decoding will remain approximately the same across benchmarks.

The decision function $f_n$ was trained prior to these evaluation as described above. During initial testing, we found that the relationship between the classifier's decision threshold $T$ and the resulting backoff percentage varied broadly across benchmarks. Thus, we calibrated our decision thresholds for each benchmark on a small subset of (input token sequence, Kl divergence) points generated from

---

[1]We will release this backoff decoding implementation shortly

the training splits of each respective benchmark, and found that the backoff percentages observed during calibration on this training set matched those seen during the test split evaluations.

All evaluations of the backoff decoding models, the speculative decoding models, and standalone $M_L$ were run on 4 A6000 GPUs. The $M_S$ evaluations were run on a single A6000.

## 4.2 INFERENCE COST-PERFORMANCE TRADE OFF

We start by demonstrating the potential inference cost savings that can be achieved with the backoff decoding framework in order to motivate its application. As mentioned, these savings directly depend on how many tokens are generated by the more efficient $M_S$ compared to larger $M_L$. We will then show the performance of the overall model at these different "backoff percentages", demonstrating what cost savings can be achieved while maintaining the performance of $M_L$, as well as what cost savings can be achieved in exchange for small performance decreases.

Figure 1 shows the average inference cost per token generation under the backoff decoding framework at different backoff percentages, compared to both $M_S$ and $M_L$ run in isolation as well as a speculative decoding benchmark. All models were prompted to generate 500 tokens in response to an open ended essay prompt, with the backoff models set to randomly backoff at the given backoff percentages. As expected, we can see that at as the backoff percentage increases, backoff decoding results in an increasingly large reduction in inference cost compared to the large model in run isolation. We also can see that at backoff percentages greater than around 80%, backoff decoding results in a greater inference cost reduction than speculative decoding, and can even achieve cost reductions of around 5-6x at high backoff percentages of around 95%.

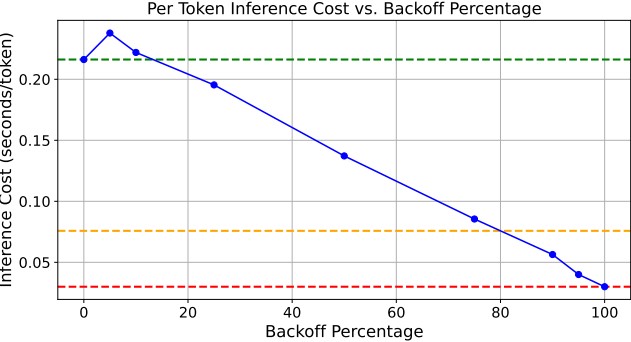

Figure 1: Inference cost per generated token vs backoff percentage

Given this impressive efficiency boost, the question now becomes how increasing backoff percentages impacts the performance of the model. Figure 2 shows the performance of our backoff decoding framework at different backoff percentages, evaluated across several benchmarks. Instead of plotting the backoff percentage directly, we have plotted the corresponding inference speeds a the given backoff percentages. We include the performance of both the oracle backoff model variant and the classifier backoff model variant. As mentioned, the oracle model cannot lead to any efficiency gains, and rather is used to demonstrate how much the performance of the backoff decoding framework could improve given a better classifier. Thus, to allow for a better comparison between the classifier and oracle performance, we have plotted the oracle results at the inference speeds the classifier would have achieved at the various backoff percentages of the oracle, following the relationship described in 1. We compare performances of both these variants to both the $M_S$ and $M_L$ in isolation, and mark the point at which backoff decoding would outperform the efficiency gains of speculative decoding. The backoff percentage at which the backoff decoding speed-ups would match those of speculative decoding was determined using the relationship in 1.

First, we note that even with this relatively simple and unrefined classifier decision function, the backoff decoding framework is able to achieve performance levels almost matching those of the standalone large model at backoff percentages of up to around 50%. On GSM8K, the classifier model even able to maintain the full performance of large model $M_L$. Furthermore, we note that the

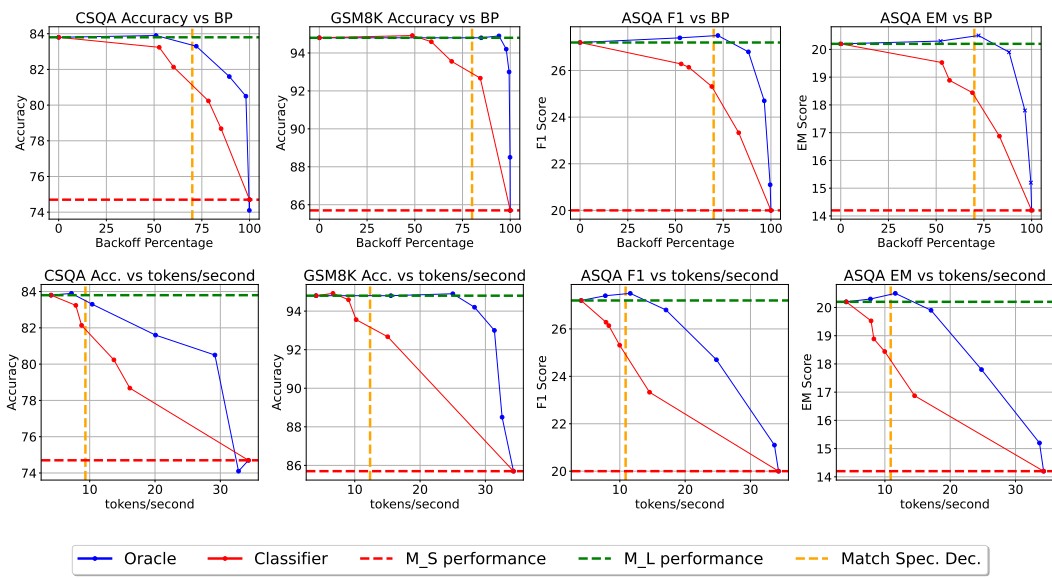

Figure 2: Benchmark performance of Oracle and Classifier Backoff models at different backoff percentages and corresponding inference speeds, compared to performance of $M_S$ and $M_L$

| KL Div. Range | Percentage of Points | | | |
| --- | --- | --- | --- | --- |
| | 0-0.5 | 0.5-1 | 1-2 | $\geq 2$ |
| CSQA | 0.73 | 0.16 | 0.09 | 0.03 |
| GSM8K | 0.94 | 0.04 | 0.01 | 0 |
| ASQA | 0.72 | 0.15 | 0.08 | 0.03 |

Table 1: KL Divergence Range for Benchmark Datasets

oracle model consistently performs better than the classifier model, and is able to back off on up to at least 70-90% of tokens with virtually no degradation of the model's performance at all (compared to $M_L$). This strong oracle performance indicates that further improvements of the classifier would result in inference cost reductions even larger than those our classifier models were able to achieve.

Beyond this, we can also see backoff decoding can maintain a majority of the performance difference between $M_L$ and $M_S$ even at very high backoff percentages for both the classifier and oracle models. This results demonstrates the tunability of our framework, giving users the option to exchange small decreases in performance for a significant further reduction in runtime - an option that other inference acceleration frameworks do not offer.

Another interesting observation to note here is that the benchmarks seem to have different sensitivities to backoff generations. For instance, the performance on CSQA and ASQA degrades much quicker as we increase the backoff percentage than it does for GSM8K, both for the oracle and classifier models. The reason for this lies in the differences in the underlying distribution of KL divergences between models across these different datasets. This can be seen in table 1, which details the distribution of KL divergences from over a small subset of points sampled from our benchmark evaluations.

In table 1, we can see that for GSM8K (which is the dataset for which we are able to back off more generations to $M_S$ without seeing a drop in performance) the distribution of KL divergences is much more skewed towards lower divergences. Thus, it seems that the performance-efficiency tradeoff of our method is dependent to how close the distributions of $M_S$ tend to be to those of $M_L$ on the given dataset or generation. The more the two models tend to diverge, the larger impact there will be on performance by backing generation off to $M_S$. This same concern in known to also apply to

speculative decoding, where greater divergences between the draft model and the generating model result in significantly less efficiency gains.

### 4.3 IMPORTANCE OF HIGH KL DIVERGENCE TOKENS

The performance of the classifier and oracle models in Figure 2 suggests that only a very small subset of all generated tokens account for a majority the performance difference between $M_S$ and $M_L$, and that the KL divergence (or an estimation of it) seems to be an effective way to identify these generations.

To test this theory, we introduce another backoff model variant, the **flipped oracle backoff model**. This variant is a modified version of the oracle model with a flipped decision function, routing all high Kl divergence generations to $M_S$ and all low KL divergence generations to $M_L$. This flipped oracle decision function tests the importance of these high Kl divergence generations by selectively routing only these generations to the weaker $M_S$. If generating these tokens with $M_L$ is vital to the quality of the overall generation, then this routing should noticeably degrade the performance of the model.

Surprisingly, it seems a very small subset of high KL divergence tokens has an overwhelming impact on the overall quality of the generation, even noticeable despite the relatively small performance differences between $M_S$ and $M_L$. Backing off on even just 0.07% of the highest KL divergence generations (Kl div. $\geq 10$) noticeably degrades the performance of the model, dropping it lower than the performance of a regular oracle backing off on the lowest 89% of its generated tokens. This result illustrates the importance of the decision function in routing generations correctly between the two models, as it shows that even just a few falsely backed off tokens can drastically degrade model performance.

| Model Type | Backoff Decision Criteria | CSQA | | ASQA | | |
|---|---|---|---|---|---|---|
| | | Acc. (%) | B-Off % | QA-F1 | QA-EM | B-Off % |
| $M_S$ | - | 63.7 | 100 | 7.4 | 4.4 | 100 |
| $M_L$ | - | 68.3 | 0 | 16.2 | 10.8 | 0 |
| Reg. Oracle | KL Div $\leq 1$ | 69.0 | 89.0 | 15.7 | 10.5 | 84.0 |
| Reg. Oracle | KL Div $\leq 2$ | 68.1 | 97.0 | 12.3 | 8.0 | 93.0 |
| Reg. Oracle | KL Div $\leq 5$ | 65.8 | 99.7 | 8.8 | 5.5 | 99.25 |
| Reg. Oracle | KL Div $\leq 10$ | 62.7 | 99.996 | 7.3 | 4.5 | 99.97 |
| Flipped Oracle | KL Div $\geq 5$ | 67.2 | 0.98 | 13.9 | 9.0 | 0.98 |
| Flipped Oracle | KL Div $\geq 10$ | 68.8 | 0.07 | 15.1 | 10.0 | 0.068 |

Table 2: Performance of flipped oracle compared to regular oracle.

### 4.4 CLASSIFIER PERFORMANCE ANALYSIS

The high performance cost incurred by incorrectly backing off on high KL divergence tokens suggests that performance of the classifier (or other decision function) in allocating generations between models correctly has a direct impact on the backoff models overall performance. Thus, we analyze the performance of the classifier used in our evaluations in order to explain the performance difference between the classifier models and the optimal oracle. By doing so, we hopefully outline some key considerations for the development of better performing classifiers (and thereby backoff decoding models) in future works.

We start by looking at the performance of the classifier on high KL divergence points at different backoff percentages. To do this, we generate a small dataset of (input sequence, KL divergence) points from the training split of the benchmarks we used above and evaluated the accuracy of our

classifier on points in three high KL divergence groups at different backoff percentages. We did not include the KL div $\geq 5$ group for GSM8K, since the dataset did not have enough points with KL div. $\geq 5$ in the subset we sampled.

The trends in accuracy on points with high KL divergence is shown in figure 3. We can see that, as we push the classifier to higher backoff percentages, the performance on these high KL div. points suffers drastically, even to the point that the classifier is labeling more points in this group incorrectly than correctly. While the decrease in accuracy is expected given that we are changing the decision threshold to manipulate the backoff percentage, we note all 3 accuracies seem to degrade equally as the backoff percentage increases. This is somewhat unexpected, as we would expect the accuracies on the higher KL divergence groups to degrade less than lower groups, since their Kl divergence values are further away from the trained classification threshold and therefore should be more confidently classified. The absence of this trend indicates that the classifier is struggling to learn a feature representation of the KL divergence and thereby isn't capturing a sense of the magnitude of the KL divergence in the points it is classifier. We note that this may be a result of the classifier being trained in a binary classification setting, since training in this way gives the classifier no sense of the magnitude of the points in the two classes. (a point with KL div. 0.51 and a point with KL div. 10 will appear identical to the classifier). This, in combination with the results in table 2, may also explain why the classifier underperformed in comparison to the oracle: if incorrectly backing off on even less than 1% of the highest KL divergence tokens degrades performance, then its no surprise that a model that will incorrectly back off on 50% of these tokens achieves weaker performance. As such, it seems that a key to pushing the classifier model's performance closer to that or the oracle model is training the classifiers to be better at classifying these high KL divergence points.

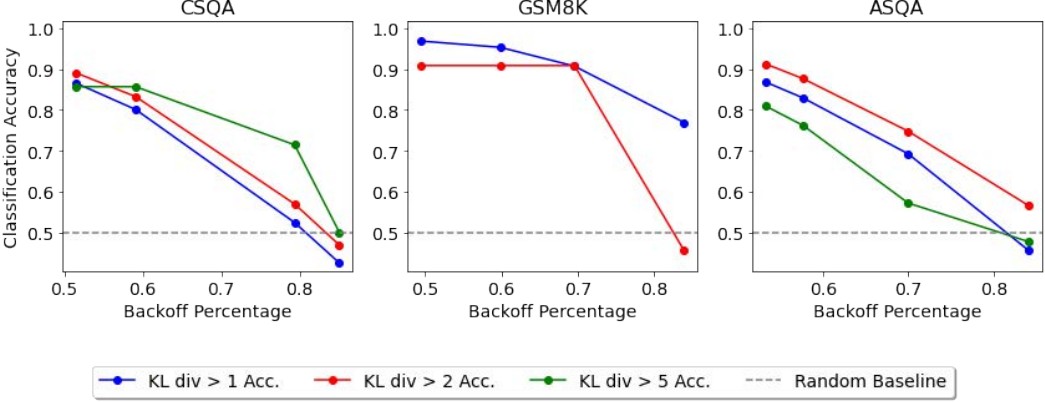

Figure 3: Accuracy on high KL divergence points across different benchmark datasets

Next, we look at how the performance of the classifier changes depending on the depth of the hidden state of $M_S$ used as input for the MLP. Recall that our decision function $f_n$ consists of an MLP defined over the nth hidden state of $M_S$. During training, we generally found that classifiers defined over deeper layers of $M_S$ performed better than those performed than those defined over shallower layers. This is illustrated in figure 4, which shows the validation accuracy of a linear layer classifier defined over layers of $M_S$ at different depths during wikitext pretraining training. While there were a few exceptions in which intermediate layers performed better than deeper layers, this trend was generally observed to be consistent across dataset types, classifier types, and training lengths. Thus, it seems that the features most suitable for KL divergence classification are extracted by $M_S$ throughout the forward pass forward pass .

Given this analysis of the performance of our classifier, we suggest a few major classifier improvements that we believe may lead to substantial classification performance improvements. Firstly, we suggest framing the optimization of the classifier such that there is a sense of KL divergence magnitude incorporated during training, as this would hopefully improve the performance of the classifier on points with high KL divergences. One simple way to do this would be to frame the optimization as a regression instead of a binary classification, as this would likely give the classifier a sense of scale regarding the KL divergences sees during training. Another way to do this might

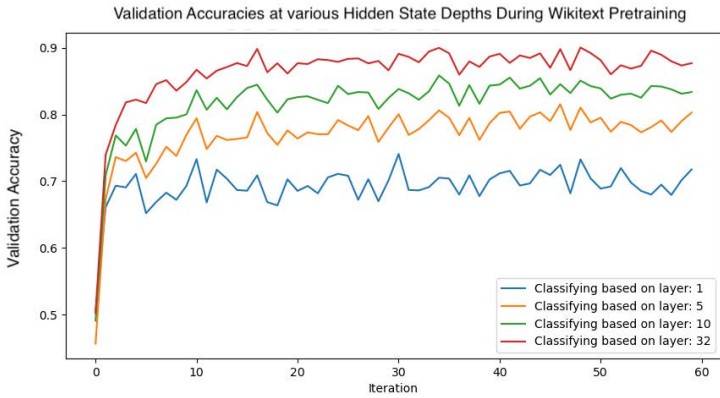

Figure 4: Validation Accuracy of Linear Classifier at Difference Layer Depths of $M_S$

be a multi-class classification, for similar reasons. Secondly, another approach worth investigating is to finetune an entire small language model on this classification task. While this would increase inference costs associated with classification, given the much stronger performance at later layers in the model its possible that this would drastically improve the models ability to learn generalize features which it can use during classification.

## 5 DISCUSSION

We presented **backoff decoding**, a inference acceleration technique for language models that allocated individual token generations between differently sized models. We've demonstrated how this framework is able to significantly accelerate language model inference depending on how many generation it allocates to a small model, and observed that even if a significant portion of generations are routed to $M_S$, most if not all of the performance of the larger model can be maintained. We also demonstrate how this framework can be used to exchange small decreases in performance for even greater reductions in inference cost, which current inference acceleration approaches do not allow. Finally, we propose that this framework is able to maintain this high performance because only a very small subset of tokens truly determine the performance differences between two model sizes, with most of the tokens having a minimal impact on the final generation quality.

The unique benefits of backoff decoding position it for several applications that are currently underserved by existing inference acceleration techniques. Most importantly, backoff decoding lets users tune the inference cost-performance tradeoff of their model, adding a flexibility to inference acceleration that currently does not exist. We imagine this to be highly useful in applications where low inference costs may become imperative only during certain times (high traffic, low compute availability, etc.), allowing users to tune their model for lower runtimes during these select windows without having to use a smaller, less performative model for all generations. Furthermore, the flexibility of our approach allows for considerable modifications to suit the framework to specific use cases. We image scenarios in which the decision function can be optimized for much more complex objectives and work off of a wide set of inputs, tuning the allocation of generations precisely to the needs of a given application.

However, there are a few limitations of our method. Firstly, to efficiently implement backoff decoding, both $M_S$ and $M_L$ typically need to be kept in RAM, such that they are quickly accessible in the case that $f_d$ routes a generation to them. This significantly increases the hardware requirements to efficiently run backoff decoding. Furthermore, while the models themselves don't need to be finetuned or trained, backoff decoding does usually require the training of the decision function $f_d$. Depending on the objective of this function, this may be difficult to optimize correctly, especially since weak classifier will drastically degrade the overall model's performance. It should also be noted that these decision functions must be kept very efficient, since their runtime will subtract from the efficiency gains realized by our method.

These findings itself to several direction of future research which we believe can greatly extend the performance of the backoff decoding framework. Mostly notably, these include the development of better decision functions, both in terms of structure and training, as well as in terms of objective. Our setup of estimating the KL divergence with neural classifiers defined over the hidden states of $M_S$ is somewhat simple, and we believe that significant improvements beyond our results can be realized if this approach were to be improved. Finally, another direction to pursue is extending our framework to include more than two models and studying whether this leads to performance and efficiency improvements, beyond those we've shown.

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
