# OpenReview forum: "Backoff Decoding: A Language Model Inference Acceleration Framework with a Tunable Efficiency-Performance Tradeoff"
_ICLR.cc/2025/Conference — ICLR 2025 Conference Withdrawn Submission_

### Official Review · Reviewer_jk23 · 2024-11-02

**Soundness:** 2
**Presentation:** 3
**Contribution:** 2
**Rating:** 5
**Confidence:** 4

**Summary:**

This paper proposes a novel generation strategy called backoff decoding that aims to improve efficiency of the inference by re-routing per-tokens generations between smaller and larger LLM depending on the difficulty of the token to be predicted. The core idea is to approximate the difficulty using the divergence between distributions induced by small and large model. Authors empirically show that this approximation works very well in practice, and further propose a model that can estimate this divergence without running the large model.

Authors performed experiments on multiple tasks using the proposed method showing that they could achieve substantial speed-ups with very minor performance degradation. Authors also provide connections to related work such as speculative sampling by highlighting its similarities and differences.

**Strengths:**

* This method provides a practical solution to achieve low-latency inference with very large LLMs.
* The method involves the original decision function that is defined so that there is a big room for further research improving the design of that component that will lead to lower approximation error.
* Experiments are clearly written and are easy to follow.

**Weaknesses:**

* Real-world results might be less predictable than the experiments presented in the paper. For instance, current LLMs are trained with trillions of tokens, which implies that the decision function might need much more training data in order to cover enough amount of text.
* There is no comparison with the speculative decoding in terms of quality vs. speed up. In contrast to this approach, speculative sampling is guaranteed to sample from the target model distribution thus does not have quality degradation. Authors made verbal connection with this related work and compared the speed-ups under oracle decision function, but it really needs an experiment comparing the speed and quality.

**Questions:**

* Following the weakness point above, why there is no experiment comparing with speculative decoding w.r.t. both speed and quality? I suspect this will be a question every reader will have if this is published at the conference. And if speculative decoding is faster, then why would we want to use back off decoding and risk performance degradation? These days speculative decoding advanced into using MLP based and ngram based speculators which are much faster than small LLMs. I suspect that under such comparison this method will be slower and show quality degradation. Am I correct that Fig. 2 shows how backoff decoding is always worse than speculative sampling assuming that speculative sampling should achieve performance of the large model?
* How would one choose the backoff threshold parameter in real life? Imagine that we want to get a solution that works the best in general setting (i.e. all domains) and its not feasible to tune T for each input we are getting. What shall we do? This feels like a major question to answer.

---

> ### Author Response · Authors · 2024-11-27
> **Full response to reviewer jk23**
>
> Thank you for your comments and questions! To address your identified weaknesses:
>
> Weakness 1: Less predictable real world results due to potentially high training data requirement for classifier training
> During our experiments, we found that the performance of the classifier decision function converged very quickly, ever after just small datasets of around 600K (prefix, kl div on next token) points. While our classifier setup was very simple, this is a strong indication that the KL divergence task doesn’t require large amounts of data.
>
> Weakness 2: No direct comparison to speculative decoding (SD)
> Regarding your point on having no direct comparison between backoff sampling and SD, we have since done this direct comparison and included the results in the Figure 2 of our resubmission. We observe that the speed ups achieved by SD vary noticeably across the benchmarks, and that the constant SD speed up we were using in the original Figure 2 was too high for most benchmarks, making the performance and efficiency of backoff sampling much more comparable to that of SD. We can see in the updated Figure 2 that for all benchmarks, our approach is able to be at or close to the performance level of M_L at speed ups at or above those of SD.
>
> To specifically answer your questions:
> 1. As mentioned above, we have included direct comparisons to SD in our resubmitted results. Regarding the second part of your question - the major advantage of backoff sampling over SD is that it is flexible enough to allow for speed-ups beyond the fixed speed-up that can be achieved with SD. Once you reach a certain optimal hyperparameter setup with SD, you have no option to push runtime even lower at the cost of some performance, which is possible through the threshold setting of backoff sampling. This means that our approach can achieve speed-ups beyond those of SD, while sacrificing some performance, allowing for the user to tune the performance-efficiency trade off of the generation to the specifications of the task. SD does not allow this tunability.
> 2. This is a very valid point, although this hyperparameter selection is an issue for all methods of this sort. The speed-ups of SD for instance rely on the choice of n, the number of tokens consecutively generated by the draft model, with the optimal n being entirely dependent on the domain in which the generation is taking place. The same can be said for other traditional generation hyperparameters, such as temperature, etc. Furthermore, in order to train the classifier our method only requires input text and no ground-truth labels (since the kl divergence between the models at each token position will become the labels). This makes training the procedure on new domains much easier than traditional domain finetuned methods. Thus, we don’t necessarily see this as a negative specific to backoff decoding compared to other inference acceleration methods like SD, although you are completely correct to note that this threshold is somewhat specific to the domain.

---

### Official Review · Reviewer_BMAn · 2024-11-03

**Soundness:** 3
**Presentation:** 3
**Contribution:** 2
**Rating:** 3
**Confidence:** 4

**Summary:**

This paper proposes an approach to reduce inference cost for larger language models (LLM). The proposed approach backs off to a smaller LM for decoding when the KL divergence between the large and smaller LMs are not too high. The decision for backing-off is made by a classifier trained to predict whether the KL divergence is above or below a threshold. The paper observes that in most cases, the divergence between small and large models are relatively small, hence this back-off strategy effectively reduces cost without hurting much quality.

**Strengths:**

This paper made interesting observations that on the tasks being considered (GSQA, GSM8K, ASQA), the KL divergence between large and smaller models are mostly very small. This insight justifies utilizing smaller models to handle a majority of decoding steps.

**Weaknesses:**

1. The proposed approach did not show superior performance in comparison to existing and similar techniques like speculative decoding. 2. The experimental results are not substantial enough to support the proposal: Only three tasks (GSQA, GSM8K, ASQA, as shown in Fig. 2) are considered, and only a single large-small model combination (Llama 70B, 8B) was experimented.
3. Although the proposed approach claims that it does not require finetuning the models itself (line 114), it nevertheless requires training a dedicated classifier to make it work. What is more, this classifier depends on the choice of the smaller LM, making this approach difficult to generalize. Lastly, the cost of running this classifier at each step did not seem to be accounted for in this paper.
4. The classifier decision threshold needs to be calibrated for each task, again making this approach difficult to use and generalize.

**Questions:**

1. All findings and experimental results in this paper came from the single choice of language model pair (Llama 70B, 8B), I wonder whether these observations and analysis still holds for more a broader range of models (e.g. Gemma).

2. It seem the proposed approach has no particular advantage over speculative decoding, as can be seen from Fig 1 and 2: Both performance and inference cost matches at around 80% backoff ratio, which means that the same cost allocated to speculative decoding can achieve about the same quality.

3. The paper claims that "Backing off on even just 0.07% of the highest KL divergence generations (Kl div. ≥ 10) noticeably degrades the performance" (line 382-383), however it seems from Table 2, at 0.07% back-off percentage, the quality didn't degrade at all (68.8 vs. 68.3 with full M_L)?

4. What is the size and cost overhead of the MLP used by the classifier? The paper did not seem to mention these info.

5. In the leftmost chart in Fig. 2, when backoff percentage is 100%, the M_S performance should be the same as Oracle, but this did not seem to be the case?

6. The paper should have provided some analysis over the quality degradation as the generation lengths increases, as the longer the generation, the less accurate smaller models will be. The KL divergence between large and smaller models should be a function of decoding step time.

7. Typos:
- Line 110: "users to not" => "users do not"
- Line 151: "achieves achieves" => "achieves"
- Table 1: "<=2" => ">=2"

---

> ### Author Response · Authors · 2024-11-27
> **Full response to reviewer BMAn**
>
> Thank you for your comments and questions! To address your identified weaknesses:
> Weakness 1: Inferior performance to speculative decoding and lacking evidence to support backoff decoding proposal
> While backoff decoding does not show superior benchmark performance (Accuracy, F1, etc.) over SD, it does show the ability to achieve inference costs lower than those of SD at a small degradation of benchmark performance. On all benchmarks, backoff sampling is able to achieve an inference speed faster than SD, allowing the user the flexibility to sacrifice performance to achieve these faster inference speeds - a flexibility that SD does not allow for. We would also like to note that we have updated the SD speed ups in figure 2 to be specific to each of the benchmarks, and that under these more accurate SD speed ups, the benchmark performance of backoff sampling almost matches that of SD.
>
> Regarding adding further benchmarks and model families: We did complete evaluations similar to those in the paper on Llama-2 13B and 7B models during preliminary testing and observed the same trends we saw in the Llama-3 results. However, we were unable to add another model family or more benchmark to our results due to resource constraints. Each benchmark was already very runtime intensive even just on the single model combination, and so it was infeasible to extend the results in this was in time for the original submission or during the rebuttal period. We will be sure to do this for future developments of our work.
>
> Weakness 2: Requirement of finetuning of classifier to effectively route generations
> Your concern about the current classifier setup lacking generalizability is valid and was definitely an area we were leaving to be further developed in future works. Our objective with this paper was more so to introduce backoff sampling as a framework as opposed to introducing the best KL divergence classifier setup, and thus we didn’t focus extensively on testing different classifier architectures and training procedures. Thus, we only used a very simple linear and MLP classifier, which although it lacked some performance and generalizability, had negligible overhead runtime costs. This overhead costs was also already factored in to runtime results.
>
> Weakness 3: Requirement to calibrate classifier for each domain/task
> While the classifier threshold did need to be calibrated to each domain specifically, this calibration could typically be done on an extremely small dataset, making calibrating the threshold pretty straightforward.
>
> To answer your questions specifically:
> 1. During preliminary testing, we also did similar evaluations on Llama-2 13B and Llama-2 7B models and found virtually identical results, indicating the framework works across models. Due to runtime constraints, we unfortunately were not able to test a separate model family like Gemma, although this is something we are hoping to do in future works.
> 2. The major advantage of backoff sampling over traditional methods like SD is that it is flexible enough to allow for speed-ups beyond the fixed speed-up that can be achieved with SD. Once you reach a certain optimal hyperparameter setup with SD, you have no option to push runtime even lower at the cost of some performance, which is possible through the threshold setting of backoff sampling. This means that our approach can achieve speed-ups beyond those of SD, while sacrificing some performance, allowing for the user to tune the performance-efficiency trade off of the generation to the specifications of the task. SD does not allow this tunability.
> 3. Thank you for pointing that out - this was a typo. The claim that we meant to make was referring to the line line 402 (flipped oracle, backing off on all points with KL div > 5), where we can see that even generating just <1% of all tokens with M_S leads to a percentage point drop in CSQA performance (68.3 to 67.2).
> 4. We tested both single linear layers, as well as 1-2 layer MLPs. The classifier we used to generate the results was a 1 layer MLP, with a hidden layer size of 1024. (4096 -> 1024 -> 2). It should also be noted that the overhead classifier cost is already included in the runtime measurements in the results.
> 5. To generate our results, we used sample-based decoding, both for the backoff sampling framework as well as M_S and M_L. Thus, there was always some variation across trials, likely explaining the difference you mention. Functionally, an oracle with 100% backoff is just sampling every token from M_S, so the performance is in theory identical.
> 6. This is a valid point - the only caveat to mention is that M_S must not necessarily be less accurate at approximating M_L as the generation length increases. Since even very long generations will still regularly rely on M_L to generate tokens, it’s possible that these few M_L tokens almost act to steer the M_S generation, correcting M_S when it starts veering away from what M_L would have generated.

---

> ### Comment · Reviewer_BMAn · 2024-12-03
>
> I'd like to thank the authors for your response. I will keep my original rating.

---

### Official Review · Reviewer_7ZCR · 2024-11-04

**Soundness:** 2
**Presentation:** 3
**Contribution:** 2
**Rating:** 3
**Confidence:** 4

**Summary:**

This paper proposes backoff decoding for inference acceleration of LLMs. Speculative decoding generates draft tokens from a small model and verifies these tokens by the large model. The authors believe that the validation is not necessary, and if the KL divergence of the next token of two models is below a threshold, the result of the small model can be adopted directly. In order not to calculate the KL divergence  every time, backoff decoding trains a classifier for determining whether the next token uses the token from small model or from the large model.

**Strengths:**

1. The trade-off between acceleration and accuracy in Backoff decoding is adjustable, making it adaptable to various scenarios.

2. The analysis on predicting KL divergence using a classifier is particularly interesting.

**Weaknesses:**

1. A large portion of the content overlaps with speculative decoding, including parts of the introduction, lines 143-155 in the methods section, and the whole Section 3.2.

2. The experimental setup is insufficiently detailed, lacking specifications on hardware. Additionally, while the authors present accuracy and inference cost results relative to the Backoff percentage, they do not visually display the trade-off between accuracy and inference cost in a single figure, which would be more intuitive.

3. The assumption that speculative decoding achieves consistent speedups across different benchmarks is questionable. SpecBench[1] shows that decoding speeds can vary by domain. The authors only used a few benchmarks; running speculative decoding on these additional benchmarks does not seem particularly costly.

4. The figures in the paper are not in vector format."

5. The 5-6x speedup mentioned in the abstract is exaggerated. Achieving a 5-6x speedup requires a Backoff percentage of 95%, at which point the performance degradation is not a 'small reduction'—it is almost equivalent to the performance of a smaller model. If 'small reductions' refer to absolute performance changes, the performance gap between the large and small models on the selected benchmarks is inherently small, making it reasonable to simply use the smaller model.

[1]  Xia, Heming, et al. "Unlocking efficiency in large language model inference: A comprehensive survey of speculative decoding." arXiv preprint arXiv:2401.07851 (2024).

**Questions:**

1. A large portion of the paper’s content overlaps with speculative decoding. Why not introduce Backoff decoding from speculative decoding? This would create a more cohesive flow, making it easier for readers to understand and would also save considerable space.

2. In Figure 1, why does the initial cost increase? Is it because the proportion of token acceptance is insufficient to cover the model switching cost and the classifier computation cost?

3. In Table 1, should the '<=2' actually be '>=2'?

---

> ### Author Response · Authors · 2024-11-27
> **Full response to reviewer 7ZCR**
>
> Thank you for your comments and questions! To address your identified weaknesses:
>
> Weakness 1: Significant overlap of backoff decoding with speculative decoding (SD)
> Even though the two methods are functionally similar in the sense that they both selectively generate tokens with two different models, we opted to introduce backoff sampling separately from SD because the two approaches have several fundamental differences that we wanted to highlight. These include:
> 1. The process for allocating tokens between the models is fundamentally different in backoff sampling compared to SD. Most notably, backoff sampling doesn’t have a rejection/acceptance procedure, and thus truly is allocating token generations between two models as opposed to using one model to suggest tokens for the other. This makes backoff sampling a true ensemble generation strategy, whereas SD is more of an inference acceleration strategy.
> 2. Due to this token allocation procedure, backoff sampling is a general framework that can be extended beyond the specific implementation we tested in the paper (which was a lot more similar to SD). For instance, the decision function that allocates generations can be arbitrarily changed, as well as the number of models being routed to. Introducing the backoff sampling out of SD would somewhat limit this generality by framing backoff sampling as a variant of SD.
> 3. Finally, SD and backoff sampling both have slightly different objectives. The objective of SD is to sample tokens from the M_L distribution at minimal runtime cost. The objective of backoff sampling is to learn to differentiate token generations based on how much the overall generation quality would be impacted if the given token was generated by an approximation model, and use this differentiation to optimize inference costs.
> Given these differences, we found it would be clearer to introduce backoff sampling separately from SD and build it up from scratch, even if parts of this build up overlapped with SD.
>
> Weakness 2: Lacking experiment and hardware details
> The SD, backoff sampling, and M_L results were collected on 4 A6000 GPUs, while the M_S results were run on a single A6000. We have included these details in the updated submission in section 4.1. We have also updated Figure 2 to show the tradeoff between inference cost and accuracy on the benchmarks instead of just backoff percentage and accuracy as you suggested.
>
> Weakness 3: Speculative decoding speed-ups vary across benchmarks
> You are completely correct in pointing out that SD does not lead to consistent speed-ups across benchmarks, and that comparing our method to SD on a per-benchmark basis would allow for much better comparison. We have since run SD on the benchmarks tested, and have updated Figure 2 to include the benchmark specific SD speedups. It turns out that the constant SD speed-up we used to compare to our method was actually higher than true speed up SD on most datasets, and that backoff decoding now gets close to matching SD performance across all datasets, while still allowing for the flexibility to achieve inference speed ups beyond those of SD in exchange for performance losses.
>
> Weakness 4: Figure formats
> These have been updated.
>
> Weakness 5: Incorrect claim that backoff decoding can lead to 5-6x speed up with only a little reduction in model performance.
> We have corrected the speed up number to be lower and have updated the wording as you suggested. We were initially referring to the theoretical speed up that the oracle model, which is able to maintain close to all M_L performance on certain datasets (GSM8K) even at 95% backoff, but given that the oracle cannot practically achieve this speed up, this statement is misleading, so we have changed it.
>
> To address your questions:
> 1. Addressed above.
> 2. While it is possible that this increase was partly caused by the classifier switching cost, this is unlikely, since these costs were typically almost negligible. The more likely explanation is that at these very low backoff percentages, the classifier is routing almost every generation to M_L, meaning that almost every token now incurs the cost of both a M_S and a M_L forward pass.
> 3. Yes, thank you. We have corrected the typo.

---

### Note · Authors · 2025-01-22

I have read and agree with the venue's withdrawal policy on behalf of myself and my co-authors.